# The Host-Plant Origin Affects the Morphological Traits and the Reproductive Behavior of the Aphid Parasitoid *Aphelinus mali*

Ainara Peñalver-Cruz [1,2], Bruno Jaloux [2] and Blas Lavandero [1,*]

1   Laboratorio de Control Biológico, Instituto de Ciencias Biológicas, Universidad de Talca, Talca 3465548, Chile; ainara.penalver@agrocampus-ouest.fr
2   IGEPP, INRAE, Institut Agro, Univ Rennes, 49045 Angers, France; bruno.jaloux@agrocampus-ouest.fr
*   Correspondence: blavandero@utalca.cl

**Abstract:** Diversifying agroecosystems through habitat management inside or outside production fields can provide alternative hosts and/or prey for natural enemies. In semi-natural habitats, parasitoids may find alternative host-plant complexes (HPC) that could allow their development when pest hosts are scarce in the field. However, morphological and physiological differences between alternative and targeted HPCs could affect the preference and fitness of the parasitoids, possibly altering their efficacy in regulating pests. In the present study, we examined two *Aphelinus mali* parasitoid populations developing on *Eriosoma lanigerum* from two host plants (*Malus domestica*-apple trees and *Pyracantha coccinea*). We hypothesized that *A. mali* from both HPCs will show different life history traits and behaviors because primary and alternative host-plants are known to induce variations in parasitoid biological performance. Our findings indicate that *A. mali* originating from *E. lanigerum* on *P. coccinea* parasitized more aphids and are smaller than those originating from *E. lanigerum* on apple. Furthermore, these parasitoids did not significantly vary their ability to attack and oviposit apple *E. lanigerum*, suggesting that *P. coccinea* could function as a suitable banker plant for *A. mali*. We discuss the potential use of *P. coccinea* in conservation biological control of *E. lanigerum* in apple orchards.

**Keywords:** biological control; aphid parasitoids; refuges; alternative hosts



## 1. Introduction

Agricultural landscapes composed of semi-natural and natural habitats may provide several ecosystem services, such as insect pest regulation [1]. The plants in these habitats may offer alternative preys/hosts, shelter and essential food resources for the natural enemies of pest herbivores [2] offering refuges for overwintering and protection from crop practices such as chemical applications or pruning [3]. Agroecological management of plant diversity in crop borders and the establishment of agroecological structures in non-cultivated areas, such as hedgerows, floral strips, or weed strips, could be designed to harbor alternative host-plant complexes (HPC) for parasitoids by including aphid hosts and the host-plants on which these aphid hosts feed. In particular, such complexes in the border could harbor populations of parasitoids and allow their development when the targeted pest is scarce in the field. However, it is essential for the success of these actions that parasitoids that develop in the alternative HPC can subsequently migrate to the crop and attack the targeted pest. Many parasitoid species have an intraspecific genetic structure and restrict transfer between host species or host races [4]. One way to favor the transfer of parasitoids between alternative and targeted HPCs is to consider an alternative HPC that consist of the same host species that develop on a different plant species. Indeed, many species of phytophagous insects include distinct biotypes, each specialized on a different host plants, with no or little exchange between hosts. Therefore, although the same herbivore species may occur on a crop plant and on associated vegetation, their

strong preferences for the different host plants will avoid the movement of individuals between these host plants. This phenomenon can prevent the risk of alternative HPCs to act as reservoirs of crop pests, but still favors the movement of parasitoids between associated and crop plants. Such parasitoid-host-plant systems are potentially useful tools in the conservation biological control of the targeted pests. However, genetic, behavioral and physiological mechanisms could prevent or compromise this ideal situation. In some cases, parasitoids developing on hosts of the same species feeding on different host plants, may genetically differentiate into two parasitoid races with varying host preferences and parasitism success on the targeted pest [4,5]. Consequently, a parasitoid from the same herbivore species but which developed on an alternative plant host in an uncultivated area, may not be as successful as a parasitoid originating from the same herbivore on the targeted host plant species; thereby, compromising the efficiency of biological control. Furthermore, parasitoids may respond to familiar semiochemicals to demonstrate preferences as adult females for the HPC in which they developed, a phenomenon called host fidelity [6]. Preferences may also result from adult females choosing to exploit those hosts that may provide the best resources for their offspring (the paradigm called "mother knows best", "naïve adaptionist" or "preference-performance" hypothesis) [7–11].

The chemical composition of the host plants on which herbivores feed may affect the capital resources (i.e., the resources accumulated during early parasitoid life stages) for the larval stages of the parasitoids, thereby affecting parasitoid development, oviposition and survivorship [12]. Parasitoids, during their larval stages, feed on the capital resources provided by their host itself [13], which allows them to acquire specific nutrients (lipids, sterols and essential amino acids) that are not available in non-host resources such as nectar [14,15]. Therefore, the quality and quantities of these resources in the host may alter the development of parasitoids [16,17]. Phytophagous insects of a given species developing on different host plants may differ in biochemical composition and this could represent resources of different nutritional value for their parasitoids. As a consequence, parasitoids developing on herbivores associated with the alternative host plant in an agroecosystem may perform differently towards the same herbivore associated with the targeted crop, largely due to the chemical composition and interactions within the HPC. Based on the above observations, it is essential before implementing conservation biological control actions, to understand the influence of the HPC on parasitoid preferences, life-history traits and parasitism rates in both alternative and targeted plant hosts, and to elucidate the mechanisms behind such life-history traits.

*Aphelinus mali* (Am) Haldeman (Hymenoptera: Aphelinidae) is the main parasitoid of the woolly apple aphid *Eriosoma lanigerum* (El) Hausmann (Hemiptera: Aphididae) [18,19]. This aphid provokes huge economic losses in apple orchards as it damages the aerial parts and roots of apple trees [20–23]. *Pyracantha coccinea* M. Roem. is a common hedgerow planted in the border of apple orchards in Chile that also hosts *E. lanigerum* [24–26]. The aphid shows strong genetic differentiation between these two host-plants and geographical features [24] which ensures that *P. coccinea* is not a source of the pest in apple orchards. *Aphelinus mali* parasitizes *E. lanigerum* on apple trees but also on *P. coccinea*. However, *A. mali* does show a lack of genetic differentiation between the two host plants (apple trees and *P. coccinea*) [24]. When given a choice between aphids from both host plants, *A. mali* prefers to parasitize aphids from apple trees, independent of the HPC of origin [25]. Therefore, any preference of *A. mali* towards host biotypes is probably related to the conditioning of the adult parasitoid rather than genetically-based preferences. Furthermore, *A. mali* emerges from the *P. coccinea* HPC earlier in the season than from the apple HPC [26], which could allow an early colonization of apple orchards by *A. mali* to control *E. lanigerum*. These characteristics suggest that production systems composed of *P. coccinea* hedgerows around apple orchards may be suitable for the conservation biological control of *E. lanigerum* by *A. mali*.

The main objective of this study is to evaluate the hedgerow *P. coccinea* as a refuge for the biological control agent of *E. lanigerum* in apple orchards, *A. mali*, by assessing its

biological performance when originating from two aphid hosts feeding either on *P. coccinea* (hereafter *Pyracantha*-El-Am) or on apple trees (hereafter Apple-El-Am). We predicted that *A. mali* parasitoids emerging from aphids feeding on apple trees and *P. coccinea* may differ in body size as morphological differences are often host-plant induced [27–29]; and, since body size is one of the major traits used to measure fitness [27], we also predicted that the parasitoids that emerge from aphids feeding on apple trees and *P. coccinea* will also differ in longevity and fecundity. Furthermore, because *E. lanigerum* established on apple trees exhibits strong host-fidelity and higher survival rates than those from *P. coccinea* [25], we suggest that the apple HPC may be more suitable for the development of *A. mali*. We therefore hypothesized that *A. mali* emerging from aphids feeding on apple trees will show greater body size and fitness, and will demonstrate greater preference for apple-*E. lanigerum* than *P. coccinea*-*E. lanigerum* thereby, exhibiting greater effectiveness as biological control agent.

## 2. Materials and Methods

### 2.1. Insect and Plant Material

Bioassays were conducted using newly emerged *A. mali* females reared on *E. lanigerum* colonies collected either from apple orchards or *P. coccinea* hedgerows in the Maule region (Chile). These colonies were kept in the laboratory under controlled conditions (22 ± 2 °C, 60 ± 10% RH and 16L:8D photoperiod) at the University of Talca (Chile). Pupae of each host origin were collected and kept in separate petri dishes of 10 cm in diameter with ventilation. Petri dishes were checked daily for emerged parasitoids. After parasitoid emergence, females and males were placed in clean petri dishes for 24 h to mate. All parasitoids were allowed to feed on diluted honey (30%) during the experiments. Parasitoids for the bioassays were ≤2 days old and were used only once.

One-month-old apple seedlings were transplanted into pots of 15 cm in diameter using a 2:1 peat/vermiculite soil mixture. Two-week-old seedlings were sprayed with fungicides (first a mix of fluopyram and tebuconazole at 400 cc/400 L and 2 weeks later, tebuconazole alone at 40 cc/100 L) to avoid fungal infections on the plants during the experiment. Plants were placed in a growth chamber (22 ± 2 °C; 16L:8D photoperiod) for 3 months. They were watered daily and a fertilizer rich in free amino acids (Terra-Sorb foliar, Bioiberica) was applied once at 200 mL per 100 L of water two weeks before aphid infestation. After this time, the apple plants were transferred to a greenhouse (Tmax: 41 °C; Tmin: 11 °C) and infested with ten third to fourth instar *E. lanigerum* from the unparasitized colony on apple trees kept under greenhouse conditions at the University of Talca. Leaves were cleaned manually with water and without disturbing aphids feeding on the stem twice a week to prevent the attack of *Tetranychus urticae* in the greenhouse. Four-month-old Gala apple trees previously infested with *E. lanigerum* in a greenhouse were used for the fecundity and parasitism bioassay.

### 2.2. Realized Fecundity and Parasitism Rate

A mated *A. mali* female emerged from aphids feeding either on apple trees (Apple-El-Am) or feeding on *P. coccinea* (*Pyracantha*-El-Am) was introduced to a mylar cage (45 × 15 cm: Height × Diameter) with a four-month old apple plant containing a 1 cm long colony of *E. lanigerum* (approximately 100 individuals per colony). Plants were renewed every day with a new aphid colony, until the death of the parasitoid. The pupae produced during the parasitoids lifespan (realized fecundity) [30] and the number of parasitoids emerging (fertility) [31] were counted. Plants with colonies already exposed to the parasitoid were kept in a growth chamber (22 ± 2 °C; 60 ± 10% RH and 16L:8D photoperiod) for 10 days after which colonies were evaluated for the number of aphids and pupae to calculate the percentage of parasitized aphids [formula by [32]: Parasitism rate (%) = (Number of mummies (pupae)/Number of mummies (pupae)+Number of aphids) *100]. Parasitoids were fed on either honey or water (control) before the bioassay to ensure that

there was no effect of the feeding history of the parasitoid before the test on the fecundity of both type of parasitoids. Treatments were replicated 10 times for *A. mali* of each origin.

### 2.3. Longevity and Morphometric Characteristics

The longevity bioassay was performed using mated Apple-El-Am (*n* = 22) and *Pyracantha*-El-Am (*n* = 25) females emerged from *E. lanigerum*. Newly emerged parasitoids were kept in Eppendorf tubes of 1.5 mL until they died. The tubes were opened every day for 5 s to renew the air inside and the food source (diluted honey 30%) was renewed every two days changing the cotton dipped in diluted honey. The longevity was determined as the number of days alive from the start of the experiment. In order to link the capital resources with the longevity and the morphological features of parasitoids from each host origin, each individual was stored in alcohol and the width of the thorax, the hind tibia and body length were determined using a digital camera and the software ImageJ (Rasband, W.S., ImageJ, U. S. National Institutes of Health, Bethesda, MD, USA,).

### 2.4. Foraging Behavior

Mated *A. mali* females from two hosts' origins (Apple-El-Am and *Pyracantha*-El-Am) were used for this bioassay. The experimental arena consisted of a cylinder of 1 cm in diameter and 1 cm height with a section of 1 cm of an apple branch, a single third or fourth instar *E. lanigerum* nymph and a cotton ball saturated with diluted honey (30%). A single mated female was introduced into the arena for the bioassay and allowed five minutes to settle. After this settlement period, the behavior of *A. mali* in the arena was observed for 15 min under a stereomicroscope with a cold constant light source. Two locations were considered: on the base of the arena and on the walls of the arena; and five behaviors: stinging, attacking, feeding on diluted honey, walking and stationary. The behaviors of "stinging" and "attacking" were considered as defined by Ortiz-Martínez et al. (2013) [24]. Total time, mean time and frequency for each of the behaviors and positions were recorded using the "tcltk" package of R software (R Core Team 2012). These parameters allowed the calculation of the occurrence and proportion of time spent in the different zones of the arena and time spent walking and stationary. Treatments were replicated 18 times for Apple-El-Am and 15 times for *Pyracantha*-El-Am.

### 2.5. Statistical Analysis

Raw data were checked for normality and homogeneity of variance using the Shapiro–Wilk W-test before performing the parametric test to data of all bioassays. When data did not follow the ANOVA assumptions even after data transformation, a nonparametric Mann–Whitney *U* test ($p < 0.05$) were performed (realized fecundity, duration, occurrence and duration per event for stinging, attacking and feeding behaviors). When data followed a binomial distribution and was nonparametric such as the parasitism rates, fertility, the proportion of time on the base of the arena or the proportion of occurrence on the base of the arena, a quasibinomial Generalized Linear Model was used. In addition, the survival test was performed to analyze the duration from the start of the observation to the first stinging, attacking and feeding events using Cox Model [33].

All data were analyzed using the software R v4.0.2 (The R Foundation for Statistical Computing 2020, Vienna, Austria). The package *survival* was used for the survival analysis.

## 3. Results

### 3.1. Realized Fecundity and Parasitism Rates

Parasitoids from different aphid host origins showed differences in their ovigeny and parasitism dynamics (Figure 1A,B). The parasitism rate per day showed a significant interaction between the origin of the parasitoid and the age of the parasitoid female ($\chi^2 = 4.636$; df = 1; $p = 0.031$). The number of parasitized aphids per day decreased quickly with time for Apple-El-Am (Figure 1A) whereas for *Pyracantha*-El-Am, a low parasitism was recorded during the first days after which it increased until a peak on the third day

and then, dropped again some days later (Figure 1B). Additionally, the number of days alive of emerged parasitoids showed a significant difference between parasitoid origin (W = 30317; *p* < 0.001) with *Pyracantha*-El-Am living significantly longer and even if the food was provided only before the bioassay, honey allowed *Pyracantha*-El-Am to live a day longer than Apple-El-Am (W = 30722; *p* < 0.001).

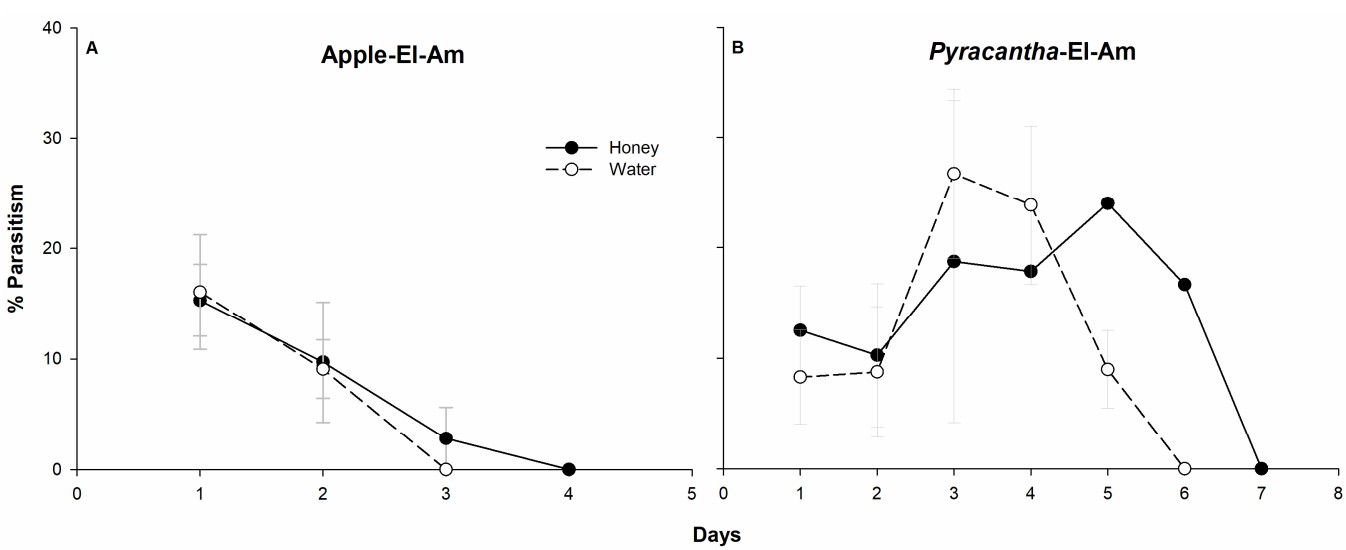

**Figure 1.** The percentage of parasitism per day by *A. mali* emerging from *E. lanigerum* of two HPC (apple trees (**A**) and *Pyracantha coccinea* (**B**)) and exposed to honey and water.

The number of pupae produced during the whole parasitoid life (realized fecundity) showed significant differences between the two parasitoid origins (W = 1000; *p* = 0.024), as *Pyracantha*-El-Am females produced significantly more pupae (3.46 ± 0.67) than Apple-El-Am (2.59 ± 0.47). There was no significant difference in lifetime realized fecundity (number of pupae produced) between both food sources (water and diluted honey) (W = 930; *p* = 0.099). On the other hand, fertility (proportion of emerged parasitoids) was similar between both parasitoid origins ($\chi^2$ = 1.713; df = 1; *p* = 0.191) and food sources ($\chi^2$ = 1.980; df = 1; *p* = 0.159).

### 3.2. Longevity and Morphological Features

The longevity of *A. mali* when given constant food supplies differed between parasitoid origins ($\chi^2$ = 36.468; *p* < 0.001) as Apple-El-Am lived twice as long as *Pyracantha*-El-Am (Figure 2A).

Apple-El-Am was larger than *Pyracantha*-El-Am (Figure 2B–D). Apple-El-Am had a longer body ($\chi^2$ = 5.338; *p* = 0.021), but the hind tibia length ($\chi^2$ = 3.494; *p* = 0.062) and the thorax width ($\chi^2$ = 2.940; *p* = 0.086) were similar between origins.

### 3.3. Foraging Behavior

Results indicate that Apple-El-Am spends more time on the base of the arena close to the aphid host ($\chi^2$ = 5.347; *p* = 0.021) compared to *Pyracantha*-El-Am (Table 1). In addition, parasitoids of both origins sting (Duration/event: W = 495; *p* = 0.745; Total duration: W = 495; *p* = 0.745; Occurrence: W = 307.5; *p* = 0.999) and attack (Duration/event: W = 429; *p* = 0.938; Total duration: W = 429; *p* = 0.938; Occurrence: W = 301.5; *p* = 0.999) their hosts similar number of times and for similar durations. Likewise, Apple-El-Am and *Pyracantha*-El-Am showed no significant difference of the duration and number of feeding events on the provided sugar source (Duration/event: W = 642; *p* = 0.102; Duration: W = 660; *p* = 0.066; Occurrence: W = 351; *p* = 0.996) (Table 1).

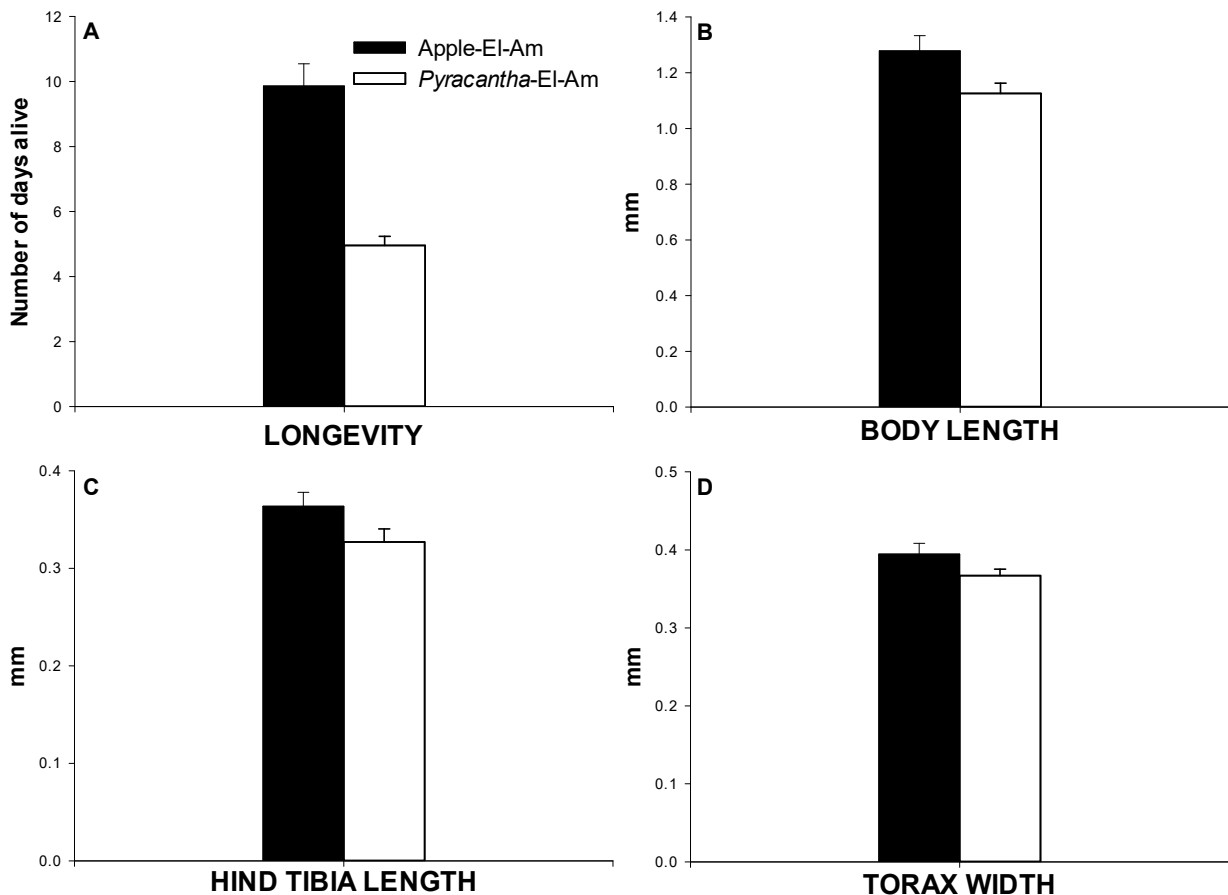

**Figure 2.** Longevity (**A**) and measurements of different morphological features (body length: (**B**); hind tibia length: (**C**); thorax width: (**D**) of *A. mali* from two aphid origins (apple trees (black bars) and *Pyracantha coccinea* (white bars)).

**Table 1.** Mean number ± SE of the occurrence, duration and duration per event of the stinging, attacking and feeding events of *Aphelinus mali* females emerging from aphids feeding on apple trees (Apple-El-Am) or on *P. coccinea* plants (*Pyracantha*-El-Am). Also, the proportion of time of *A. mali* spent at the base of the arena and moving on the base or on the walls of the arena.

| Behaviour | HPC | Stinging | Attacking | Feeding |
|---|---|---|---|---|
| Duration (seconds) | Apple-El-Am | 77.83 ± 30.01 | 13.49 ± 4.75 | 242 ± 83.39 |
| | *Pyracantha*-El-Am | 120.82 ± 50.68 | 8.43 ± 4.97 | 55.77 ± 19.84 |
| Occurrence | Apple-El-Am | 1.61 ± 0.56 | 2 ± 0.71 | 1.22 ± 0.35 |
| | *Pyracantha*-El-Am | 0.93 ± 0.40 | 1.2 ± 0.67 | 1.07 ± 0.30 |
| Duration per event (seconds) | Apple-El-Am | 41.02 ± 25.16 | 3.04 ± 0.95 | 171.49 ± 70.41 |
| | *Pyracantha*-El-Am | 100.12 ± 50.33 | 2.08 ± 0.81 | 40.81 ± 15.23 |
| **HPC** | **Prop. at base of the arena** [1] | **Prop. Moving at base of the arena** | **Prop. Moving on the walls** | |
| Apple-El-Am | 0.72 ± 0.08 * | 0.66 ± 0.08 | 0.51 ± 0.10 | |
| *Pyracantha*-El-Am | 0.44 ± 0.09 | 0.73 ± 0.06 | 0.75 ± 0.07 | |

[1],* $p < 0.05$.

*Aphelinus mali* from the two origins showed no significant difference in the time to the first sting (LogRank test = 0; *p* = 1; Figure 3A), attack (LogRank test = 0.37; *p* = 0.5; Figure 3B) and sugar source feeding (LogRank test = 2.98; *p* = 0.08; Figure 3C).

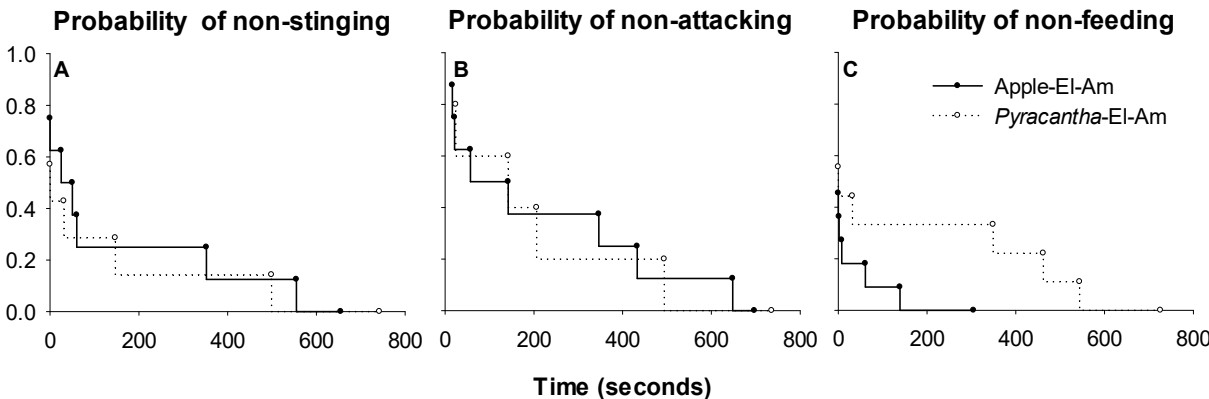

**Figure 3.** Survival tests to evaluate the probability of non-stinging (**A**), non-attacking (**B**) and non-feeding (**C**) by the parasitoid *A. mali* from different aphid host origins (apple trees or *Pyracantha coccinea*).

## 4. Discussion

Although no genetic differentiation has been detected between Apple-El-Am and *Pyracantha*-El-Am [24], the host–plant complex of origin affects a number of life history traits of the parasitoid females. Indeed, the origin of the parasitoids affects parasitism dynamics, fecundity and body morphology. Apple-El-Am females live for longer, have greater body length and remain for longer when surrounding their hosts when exploiting an Apple-El-Am HPC compared to *Pyracantha*-El-Am HPC. However, *Pyracantha*-El-Am females had greater realized fecundity, laying more eggs during their lifespan. These differences do not seem to be linked to behavioral changes or preferences, as *A. mali* from the two origins show similar frequency and duration of oviposition and attack when faced with its host *E. lanigerum* on apple trees.

One of the main differences between parasitoids from the two origins concerns the fecundity and parasitism dynamics over the female's lifetime. *Pyracantha*-El-Am females produced more pupae and over a longer period than Apple-El-Am when food was supplied only at the beginning of the assay. However, when given continuous access to food, Apple-El-Am females extended their lifetime by 50%. The ovigeny patterns were previously categorized into two groups; pro-ovigenic and synovigenic parasitoids: some parasitoids have mature eggs ready to use for parasitizing as soon as they emerge (pro-ovigenic) but others emerge with at least some immature eggs and after a few hours of preoviposition period, they continuously mature eggs during their lifespan (synovigenic) [34,35]. However, parasitoid ovigeny is not strictly defined by these two extremes as there are multiple intermediate cases specified with a continuous ovigeny index [between 0 (synovigenic) and 1 (pro-ovigenic)]. In fact, parasitoids such as *Gronotoma micromorpha* (Hymenoptera: Figitidae: Eucoilinae) have recently been defined as prosynovigenic parasitoids with two reproductive cycles during their lifetimes [36]. Jervis et al. (2008) [37] stated that parasitoid species show divergent curve types to define the relationship between the age and the realized fecundity (ovigeny index). In the present study, we found that Apple-El-Am displayed a Type 1 curve which is considered a highly pro-ovigenic parasitoid, whereas *Pyracantha*-El-Am showed a Type 2 curve indicating that this parasitoid is synovigenic (Figure S1). These characteristics blur the synovigenic status [34,35,38,39] characteristic of Aphelinids as the specific HPC determined their ovigenic pattern. Therefore, this could probably indicate that the composition of the capital resources provided by *E. lanigerum* host during parasitoid larval development were different depending on host plant and that this capital resources could affect ovigeny dynamics once adult. Indeed, food quality and quantity are known to shape the age specific fecundity curve of insects [40]. Parasitoids from *Aphelinus* genera forage plant derived food such as nectar and extrafloral nectar, but also host derived food, as honeydew or even hemolymph through host feeding [34]. Thus, *Pyracantha*-El-Am may have had access to less quality and quantity capital resources

compared to Apple-El-Am and needs external adult food sources such as host feeding to mature their eggs and consequently, start parasitizing their host more actively after a couple of days. This is also the case of the parasitoid *Itoplectis naranyae* (Hymenoptera: Ichneumonidae) that performs host feeding for egg production, associated with a delay of at least three days to mature their eggs [41]. This delay should reflect the metabolization of the nutrients, the promotion of oogenesis and the incorporation of the nutrients into the eggs [42,43]. *Pyracantha*-El-Am spent more days parasitizing before dying. This could be related to the ovigenic patterns that often determine parasitoid fecundity and longevity [35] as more synovigenic parasitoids are expected to have a longer lifespan than pro-ovigenics or less synovigenic parasitoids.

There are considerable evidences of the positive correlation between body size (measured by the hind tibia length) and the longevity and fecundity in parasitoids [44–47]. However, how this size is affected by the host origin from different host plants is relatively unknown [48]. In some instances, the body size of parasitoids appears to be unaffected by different plant genotypes as it is the case of the parasitoid *Aphidius colemani* Dalman (Hymenoptera: Braconidae) attacking aphids on different genotypes of quackgrass [48]. In the present study, Apple-El-Am showed significantly greater body length compared to *Pyracantha*-El-Am. An explanation for these morphological differences could be related to the host size as it was demonstrated that the parasitoid *Aphidius ervi* Haliday (Hymenoptera: Braconidae) with high gene flow between parasitoid origins [49,50] as *A. mali* [24], showed morphological variations associated to the host size [51]. However, as found by Ortiz-Martínez et al. (2013) [25], *E. lanigerum* aphids feeding on these two host plants have similar sizes. On the other hand, prey search, selection and feeding by natural enemies are affected by the plant structure that may interrupt aphid biology and its chemical composition (poor nutrient content and antibiotic constituents) [52]. Thus, different factors, such as antibiotic resistance [52] or trichome-mediated defenses [53], can negatively affect parasitoid and predator performance and, most likely, their morphological features. Therefore, these parasitoid body size differences may be indirectly related to the host plant as the possible reduction of capital resources for the parasitoids may alter the allocation of resources towards the soma (exoskeleton and musculature of adults) or the nonsoma (reproductive tissues and gametes) [54]. The hedgerow *P. coccinea* seems to be a suitable host for *E. lanigerum* as they normally reproduce on this plant, but the honeydew excreted by this aphid feeding on *P. coccinea* is insignificant compared to when those are feeding on apple plants (personal observation). Quality and quantity of honeydew production are known to be positively correlated [55], thus, greater honeydew production could also mean greater quality of honeydew. Therefore, *P. coccinea* may not be as nutritious as the apple trees for *E. lanigerum* which, in turn, could suggest that the capital resources for parasitoids provided by aphids feeding on *P. coccinea* are not of great quality for *A. mali* development during the larval stages and performance as adult. Future research investigating the composition of capital resources for parasitoids of different host origins is required to better understand the behavioral changes of the parasitoids and the needs for alternative host for a more successful biological control by the parasitoids that take refuge in those aphids.

Implementing a hedgerow as a conservation biological control strategy can provide agronomical, climatic and ecological benefits to the agroecosystem. Our results show the potential of *P. coccinea* hedgerows to improve the ecosystem service of the biological control of pests such as *E. lanigerum* in apple orchards through the provision of an efficient alternative host for the specialist parasitoid *A. mali*. However, the distribution of *P. coccinea* in the landscape (e.g., surrounding the orchards, only at one side of the orchards or included inside the orchard) and the interactions among natural enemies, as well as its potential to represent a source of other pests or pathogens, should be considered. All the possible benefits and issues should be addressed depending on the landscape and practice intensity, and further research should be undertaken to deliver the adequate recommendations for farmers to ultimately benefit from the ecosystem services that *P. coccinea* could provide.

## 5. Conclusions

The HPC of origin of the parasitoid has great significance for the fecundity dynamics, body size and foraging behavior of *A. mali*, but it does not affect the duration and occurrence of attacking and stinging its host *E. lanigerum*. Results of the present study confirm that *E. lanigerum* from apple trees is a great host for Apple-El-Am as it showed a greater preference for its host–plant complex of origin, live for longer and were bigger than *Pyracantha*-El-Am on apple trees, which suggests a strong host fidelity to apple-originated *E. lanigerum*. Moreover, *Pyracantha*-El-Am exhibited greater realized fecundity on *E. lanigerum* developing on apple trees, but has similar attack and oviposition behavior to Apple-El-Am. This, in turn, suggests that *A. mali* from both origins is a suitable parasitoid to control *E. lanigerum* in apple orchards. *Pyracantha coccinea* as a hedgerow adjacent to apple orchards has great potential for implementation on conservation biological control programs as it provides *A. mali* populations: (1) early in the season without spillover of *E. lanigerum* to apple orchards and (2) as efficient as Apple-El-Am to control the targeted aphid pest.

**Supplementary Materials:** The following supporting information can be downloaded at: https://www.mdpi.com/article/10.3390/agronomy12010101/s1, Figure S1. The realized fecundity per day (A) and the percentage of parasitism per day (B) during the lifespan of A. mali emerging from E. lanigerum of two HPC (apple trees and Pyracantha coccinea) and exposed to honey and water. Also, theorical patterns of age-specific realized fecundity for parasitoids explained by Jervis et al. 2008 (C).

**Author Contributions:** Conceptualization, A.P.-C. and B.L.; methodology, A.P.-C. and B.L.; formal analysis, A.P.-C., B.L. and B.J.; investigation, A.P.-C.; data curation, A.P.-C., B.L. and B.J.; writing—original draft preparation, A.P.-C., B.L. and B.J.; writing—review and editing, A.P.-C., B.L. and B.J.; project administration, A.P.-C.; funding acquisition, A.P.-C. and B.L. All authors have read and agreed to the published version of the manuscript.

**Funding:** This research was funded by the Fondo Nacional de Desarrollo Científico y Tecnológico (FONDECYT) Postdoctoral Grant number 3160233. The collaborative travel of Dr. Bruno Jaloux was funded by the FONDECYT Regular Grant number 1140632. B.L. was funded by the ANID/PIA/ACT192027 during the writing of the manuscript.

**Institutional Review Board Statement:** The animal study protocol was approved by the Institutional Review Board (or Ethics Committee) of Comité Institucional de Ética, Cuidado y Uso de Animales de Laboratorio (CIECUAL) of the Universidad de Talca, Chile (protocol code 2016-02-A and 04/01/2016).

**Data Availability Statement:** The data presented in this study are available on request from the corresponding author.

**Acknowledgments:** The authors would like to thank Cinthya Villegas and Artzai Jauregui for the assistance in the logistics of the experiments.

**Conflicts of Interest:** The authors declare no conflict of interest.

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
