# Peer review of "The Host-Plant Origin Affects the Morphological Traits and the Reproductive Behavior of the Aphid Parasitoid Aphelinus mali"

_agronomy, doi:10.3390/agronomy12010101_

Round 1
Reviewer 1 Report
The MS (Agronomy 5116662) aimed to elucidate the potential of using scarlet firethorn for conservation biological control of Aphelinus mali in apple orchards. The MS is sound, well-written and uncomplicated. As there are only minor corrections required on the attached MS, it is recommended for publication.

Reviewer 2 Report
Dear Author(s)
The purpose of this draft is to look at how well an aphid parasitoid performs on aphid-hosts that come from two different host plants (i.e., apple trees and Pyracantha), using morphometric, biological, and behavioral parameters to compare them. Furthermore, the researchers also discuss whether Pyracantha may be used for the conservation biological control program.
Overall, the work is well-written and easy to read. The findings are particularly noteworthy because they suggest that parasitoids from both origins were efficient in controlling aphids in apple orchards, regardless of their aphid-host origin. As a result, the use of a hedgerow adjacent to apple orchards has a lot of promise for conservation biological control programs.
Nonetheless, I have noticed a few points that need minor adjustments by authors before publication, which are stated and marked in the files attached.
All the best,
Reviewer
